# Farnesoid X Receptor Overexpression Decreases the Migration, Invasion and Angiogenesis of Human Bladder Cancers via AMPK Activation and Cholesterol Biosynthesis Inhibition

**DOI:** 10.3390/cancers14184398

**Published:** 2022-09-09

**Authors:** Chien-Rui Lai, Yu-Ling Tsai, Wen-Chiuan Tsai, Tzu-Min Chen, Hsin-Han Chang, Chih-Ying Changchien, Sheng-Tang Wu, Hisao-Hsien Wang, Ying Chen, Yu-Huei Lin

**Affiliations:** 1Department of Biology and Anatomy, National Defense Medical Center, Taipei 114, Taiwan; 2Department of Pathology, Tri-Service General Hospital, National Defense Medical Center, Taipei 114, Taiwan; 3Department of Internal Medicine, Tri-Service General Hospital, National Defense Medical Center, Taipei 114, Taiwan; 4Division of Urology, Department of Surgery, Tri-Service General Hospital, National Defense Medical Center, Taipei 114, Taiwan; 5Department of Urology, Cheng Hsin General Hospital, Taipei 112, Taiwan; 6Post-Baccalaureate Program in Nursing, College of Nursing, Taipei Medical University, Taipei 110, Taiwan

**Keywords:** FXR, bladder cancer, migration, invasion, angiogenesis, AMPK, cholesterol, atorvastatin

## Abstract

**Simple Summary:**

Our previous studies characterized that FXR overexpression results in the inhibition of migratory, adhesive and angiogenic abilities through the proteosome degradation pathway in human bladder cancer cells. Since cholesterol metabolism plays an important role during cancer progression, we investigated the role of cholesterol biosynthesis-related proteins expression and their signal transduction pathway in human bladder cancer cells. We confirmed FXR overexpression decreased muscle invasive human bladder cancer cell T24’s metastatic ability in nude mice animal models. Moreover, statin usage showed potent enough efficacy to strengthen the enhancement of FXR-inhibited migration, adhesion and angiogenesis in human urothelial carcinoma cells. Additionally, clinical data showed survival benefits of statin usage in stage 0–I bladder cancer patients. Our results suggested that FXR overexpression, combined with atorvastatin treatment, may provide a potential therapeutic strategy for the treatment of human urothelial carcinoma in the future.

**Abstract:**

Bladder cancer is one of the most prevailing cancers worldwide. Although treatments for urothelial carcinoma have improved, the rate of recurrence observed in the clinic is still high. The aim of this study was to evaluate whether cholesterol biosynthesis is involved in the effect of Farnesoid X Receptor (FXR) on bladder cancers. FXR overexpression contributed to activation of 5′ AMP-activated protein kinase (AMPK) and decreased cholesterol levels. FXR overexpression reduced cholesterol biosynthesis and secretion by downregulating Sterol Regulatory Element Binding Protein 2 (SREBP2) and 3-Hydroxy-3-Methylglutaryl-CoA Reductase (HMGCR) expression. In addition, an AMPK inhibitor, dorsomorphin, reversed the inhibition of migration, invasion and angiogenesis by FXR overexpression. In a metastatic xenograft animal study, FXR overexpression suppressed bladder cancer lung metastasis by decreasing matrix metalloproteinase-2 (MMP2), SREBP2 and HMGCR expression. Moreover, FXR overexpression combined with atorvastatin treatment further enhanced the downregulation of the migratory, adhesive, invasive and angiogenic properties in human urothelial carcinoma. In clinical observations, statin administration was associated with better survival rates of early-stage bladder cancer patients. Our results may provide guidance for improving therapeutic strategies for the treatment of urothelial carcinoma.

## 1. Introduction

Bladder cancer is one of the most universal cancers worldwide. Previous statistical analyses showed that smoking and exposure to chemicals, such as aromatic amines, are the main risk factors for bladder cancer [1]. Moreover, the incidence of bladder cancer is higher in men than in women [2]. In general, the 5-year survival rate of bladder cancer patients is 77%. However, the survival rate decreases significantly to 5% for those with metastatic bladder cancer [3]. The treatment for non-muscle-invasive bladder cancer (NMIBC) is transurethral resection followed by intravesical chemotherapy, and the typical treatment for muscle-invasive bladder cancer (MIBC) is radical cystectomy and neoadjuvant chemotherapy. Although the treatment of bladder cancer has improved markedly, approximately 60–70% of patients with NMIBC experience recurrence within 3 years after tumor resection [4,5]. Thus, new therapeutic strategies for bladder cancer patients are needed, and this an important issue in urothelial carcinoma research.

The farnesoid X receptor (FXR) is a bile acid-activated transcription factor and a member of the nuclear receptor superfamily [6]. Regulation of FXR (encoded by the NR1H4 gene) downstream targets contributes to glucose [7], bile acid [8], and fatty acid metabolisms [9]. FXR is also a bile acid sensor that upregulates small heterodimer partner (SHP) expression [10,11,12]. The downregulation of the expression of cholesterol 7α-hydroxylase (CYP7A1), which is the first and rate-limiting enzyme in the classic pathway of bile acid synthesis, helps to maintain cholesterol homeostasis [13]. The induction of FXR and SHP expression inhibits the CYP7A1 expression and decreases bile acid synthesis through a negative feedback mechanism related to metabolism of cholesterol [13]. Previous studies have shown increased cholesterol synthesis in cancer cells [14,15]. The enzymes involved in cholesterol biosynthesis pathways are regulated by sterol regulatory element-binding protein (SREBP), which acts as a regulator of intracellular cholesterol levels [16]. Moreover, AMP-Activated Protein Kinase (AMPK), a serine/threonine protein kinase, inactivates 3-Hydroxy-3-Methylglutaryl-CoA Reductase (HMGCR), which results in the reduction of cholesterol synthesis [17,18]. Therefore, upregulation of FXR may reduce cholesterol content in cancer cells.

Our previous studies have shown that FXR contributes to bladder cancer cell migration, invasion, and angiogenesis through the proteasomal degradation pathway [19]. In this study, we aimed to determine whether cholesterol metabolism-related signaling is involved in the induction of FXR expression in human bladder cancer cells. FXR overexpression induced AMPK phosphorylation and decreased cholesterol synthase-related protein expression; these findings may provide new insight into bladder cancer treatment in the future.

## 2. Materials and Methods

### 2.1. Cell Culture

The Tri-Service General Hospital 8301 (TSGH8301) cell line was given by the Urology division, Tri-Service General Hospital, National Defense Medical Center. The culture medium of TSGH8301 was RPMI 1640 (Thermo Fisher Scientific, Waltham, MA, USA). The T24 cell line was purchased from the Bioresource Collection and Research Center (BCRC) in Taiwan and cultured in McCoy’s 5a. Ten percent fetal bovine serum (FBS, Thermo Fisher Scientific, Waltham, MA, USA), 1% sodium pyruvate, and 1% L-glutamine (Corning, NY, USA) were added to all the media mentioned above. Human umbilical vein endothelial cells (HUVECs) were acquired from the BCRC and incubated in an endothelial cell medium (ScienCell Research Laboratories, Carlsbad, CA, USA). All these cell lines were cultured in a 5% CO_2_ incubator at 37 °C.

### 2.2. Doxycycline-Induced Overexpression System

FXR (NR1H4, NM_001206979.2) cDNA transcript variant 1 clone was provided by GenScript Biotech (Piscataway, NJ, USA). The FXR sequence was cloned into a TRE (Tetracycline response element) promoter in a tetracycline-inducible plasmid, pAS4.1w. Puro-aOn by In-Fusion Sanp Assembly Master Mix (Takara Bio, San Jose, CA, USA). Puro-aOn was used as a vector control. 293T cells were used to generate lentiviral vectors harboring FXR-pAS4.1w., Puro-aOn, pCMV-dR8.91, and pMD2.G. by Lipofectamine 3000 reagent (Thermo Fisher Scientific, Waltham, MA, USA). All experiments followed the instructions according to the manufacturer. The lentivirus package plasmids and pAS4.1w. Puro-aOn were provided by the National RNAi Core Facility at Academia Sinica in Taiwan. After puromycin (2 µg/mL) selection, a FACSAria IIIu sorter (BD, Franklin Lakes, NJ, USA) was utilized to collect the bladder cancer cells mentioned above. The mRNA and protein expression of FXR were evaluated after incubation with doxycycline for 72 h.

### 2.3. Drugs and Reagents

Dimethyl sulfoxide (DMSO), doxycycline hyclate, 3-(4,5-dimethylthiazol-2)-2,5-diphenyltetrazolium bromide (MTT), and Coomassie brilliant blue G-250 were purchased from Sigma-Aldrich (St. Louis, MO, USA). Dorsomorphin (Compound c) and atorvastatin were purchased from Sigma.

### 2.4. MTT Assays

Two different human urothelial carcinoma cells were cultured at a density of 5 × 10^3^ cells (TSGH8301) and 3 × 10^3^ cells (T24) in each well in 96-well plates. After different treatments, the MTT reagent was added and incubated for 3 h. DMSO was added and measured at 590 nm after the MTT reagent was removed.

### 2.5. Cholesterol Cell-Based Assay

After different treatments, human bladder cancer cells were seeded on coverslips and incubated overnight. The cells were rinsed with PBS and fixed. After washing with a wash buffer, Filipin III solution (Abcam, Boston, MA, USA) was added and incubated for 1 h. Then, the cells were mounted with mounting medium (Gel Mount Aqueous, Sigma), and images were captured by a digital camera Nikon D1X (Carl Zeiss, Oberkochen, Germany).

### 2.6. Total Cholesterol Level Measurement

The concentrations of total cholesterol in the conditioned media were measured by the Cholesterol assay kit (Abcam, Cambridge, UK) according to the manufacturer’s instructions. The values measured by Enzyme-Linked Immunosorbent Assay (ELISA) were corrected according to the dilution factor and expressed as µg/μL.

### 2.7. Wound-Healing Migration Assay

Human bladder cancer cells were seeded in plates. After different treatments, a 200 µL pipette tip was used to generate scratch wounds. Six hours later, the width of the wounds was photographed and analyzed by ImageJ.

### 2.8. Adhesion Assays

Fibronectin (1 µg/mL) was used to coat wells and incubated at 37 °C for 30 min. TSGH8301 and T24 cells were suspended and cultured in the precoated wells. The cells were incubated at 37 °C for 50 min. Then, PBS was used to wash and remove the nonadherent cells. The adhesive cells were fixed and stained. Three different fields in each well were selected and photographed. The numbers of adhesive cells in each field were calculated.

### 2.9. Transwell Assays

3% of Matrigel in McCoy’s 5A medium was applied to the upper chamber of Transwell (Corning Costar, Midland, NC, USA) at 37 °C for 2 h. Then, 3 × 10^4^ T24 cells were added to the upper chamber of Transwell. After incubation for 18 h, the bladder cancer cells in the lower chamber were fixed and stained. Finally, three different fields of each Transwell were randomly captured and analyzed for the invasive cells.

### 2.10. Tube Formation Assays

Precooled Matrigel (50 mL/well) was added to 96-well plates and incubated at 37 °C for 1 h. HUVECs (1 × 104) in 50% conditioned medium (CM) were seeded in each well. After incubation for 6 h, tube formation was imaged. Then, AngioTool was used to analyze and quantify the tube length and number of branch points.

### 2.11. Western Blotting

After various treatments, a protein extraction buffer (GE Healthcare Life Sciences, Chicago, IL, USA) supplemented with proteinase and phosphatase inhibitors (MedChem Express Monmo uth Jucntion, NJ, USA) was used to collect different groups of cells. All the protein samples were electrophoresed on 11% SDS–PAGEPAGE gels and then transferred to nitrocellulose membranes (NC membranes) (Bio-Rad, Berkeley, CA, USA). The membranes were cut into strips, blocked with BlockProTM1Min protein-Free Blocking Buffer (Visual protein, Taipei, Taiwan) and incubated with primary antibodies overnight at 4 °C. Tris-buffered saline with Tween 20 (TBST) was used to wash the membranes. All the strips were incubated in a 1:5000 dilution of HRP-conjugated anti-mouse or anti-rabbit IgG antibodies (Cell Signaling Technology) for 1 h at room temperature. Finally, an ECL substrate developing solution (Bio-Rad, Berkeley, CA, USA) was reacted with the blots. The signals were observed by Xplorer (SPOT Imaging, Sterling Heights, MI, USA) and were analyzed and quantified by ImageJ. The signal density of control was set to 100%, and the signal density of the test samples was expressed as values relative to that of the control sample. All the antibodies used were listed in Appendix A.

### 2.12. Xenograft Mouse Model

The animal experiments were authorized by the Laboratory Animal Center in National Defense Medical Center, Taiwan (IACUC No. 19-157). BALB/c nude mice (20–25 g) were used and bought from BioLASCO, Taiwan. During the experiments, the animals were anesthetized by an O_2_/isoflurane mixture. Then, the mice were assigned to two groups: the control (*n* = 9) and FXR-overexpressing groups (*n* = 9). A total of 1 × 10^6^ T24-Luc2 cells with and without FXR overexpression were administered via tail vein injection twice a week for three weeks. Doxycycline (2 mg/mL) was added to the drinking water with 0.1% sucrose; this water was given to the mice after the first day of tail vein injection, and the water was replenished every 2 days. An in vivo imaging system (IVIS) was used to monitor the bioluminescence density in the images every week. After 6 weeks, the animals were sacrificed, and the right lungs were harvested and fixed with 10% formalin. Then, the lung tissue weas embedded in paraffin and sectioned for routine hematoxylin and eosin (HE) and immunohistochemistry (IHC) staining. The left lungs were homogenized and sonicated for Western blotting.

### 2.13. HE and IHC Staining

The animal lungs were excised and fixed in 10% formalin. Then, lung tissues were sliced into 5-μm-thick sections. HE staining was used for histological evaluation. The expression levels of Matrix metalloproteinase-2 (MMP2), Sterol regulatory element-binding transcription factor 2 (SREBP2) and 3-hydroxy-3-methylglutaryl-Coenzyme A reductase (HMGCR) were assessed in the lung tissue by IHC staining. IHC staining was performed by a Ventana BenchMark ULTRA system (Roche, Basel, Switzerland). The primary antibodies were diluted in Antibody Dilution Buffer (Ventana). Antigen retrieval was conducted according to the manufacturer’s standard protocol. Secondary goat anti-rabbit antibodies (Jackson ImmunoResearch Laboratories, West Grove, PA, USA) were adopted. The expression of the aforementioned proteins was examined, and the images were randomly captured in each group.

### 2.14. Kaplan–Meier Analysis of Bladder Cancer Patients Administered Statins

The bladder cancer clinical patient data were received from the Taipei Medical University Institutional and Clinical Database (TMUCRD) in northern Taiwan, which contains medical records of almost 3 million patients from Taipei Medical University Hospital, Wan Fang Hospital, and Shuang Ho Hospital. The database includes demographic information, clinical information, laboratory examine results, and drug prescriptions. The data period covered by TMUCRD was between 1998 and 2020. The study was approved by the Institutional Review Board ethics committee (TMU-JIRB No.: N202205087).

A retrospective clinical-based cohort study was performed in BLCA (bladder carcinoma) patients with Stage 0–I who did or did not take statins between January 2006 to December 2020 to evaluate the effects of statin use. 1598 bladder cancer patients visited the three affiliate hospitals of TMU. After excluding patients at stages II, III, and IV and patients with other tumors, 836 BLCA patients were identified. The primary analysis of this study were the overall survival and bladder cancer-free survival. 

The index date is the date of the initial bladder cancer diagnosis, and the statin usage was defined as drugs prescribed within two years of the index date (ATC: C10AA, C10B). In this research, the Charlson Comorbidity Index (CCI) was utilized to account for comorbidities as described by the CDMF ICD-9 and ICD-10 scoring systems. The CCI is a method of accessing mortality for patients with specific comorbid conditions and is currently the most extensively used approach for controlling confounding owing to comorbidities in epidemiological research. (Detailed description shown in Table 1).

### 2.15. Statistical Analysis

All experiments were repeated 5 independent times. The results were presented as the mean ± standard error of the mean (SEM). Kruskal–Wallis test was used to analyze the differences. Mann–Whitney test was performed for post hoc analysis. In addition, Chi-squared and Student’s *t*-tests were used to compare subject characteristics in the databased section. The log-rank test was used to compare the two groups’ Kaplan–Meier survival probabilities. Multiple risk factors were assessed using the Cox proportional hazards regression model for bladder cancer patients’ all-cause and bladder cancer-related mortality. All tests were two-tailed; the alpha level of significance was set to *p* < 0.05, which was analyzed by SAS 9.3 software (SAS Institute Inc., Cary, NC, USA). Statistical significance was defined at *p* < 0.05.

## 3. Results

### 3.1. Overexpression of FXR Reduced the Cholesterol Content in TSGH8301 and T24 Cells

In previous studies [19], we developed a system for overexpressing FXR in NMIBC TSGH8301 and MIBC T24 cells. After FXR overexpression, the intracellular cholesterol content and cholesterol secretion were obviously decreased in both TSGH8301 and T24 cells (Figure 1A,B). Furthermore, the proteins related to cholesterol biosynthesis and uptake, including SREBP2 and HMGCR, were significantly decreased in TSGH8301 and T24 cells (Figure 1C). In addition, the levels of the SREBP-inhibiting protein phosphorylated 5′ AMP-activated protein kinase (p-AMPK) [20,21] were increased significantly in both TSGH8301 and T24 cells (Figure 1C).

The activation of PI3K/AKT/mTOR signaling increases intracellular cholesterol levels by increasing cholesterol synthesis via SREBP activation [22]. Western blotting showed that PI3K/AKT/mTOR signaling was significantly decreased after FXR overexpression in T24 human bladder cancer cells (Appendix A). Moreover, dorsomorphin did not rescue or additionally inhibit the expression of p-mTOR and p-Akt, while FXR was overexpressed in T24 cells (Appendix A). These results suggested that FXR overexpression may decrease cholesterol synthesis in human TSGH8301 and T24 bladder cancer cells by reducing SREBP2, HMGCR, and PI3KAKT/mTOR signaling and enhancing AMPK phosphorylation.

### 3.2. Survival Rate and Expressions of PRKAA1 (AMPK), SREBP2 and HMGCR in Bladder Cancer Patients and Bladder Cancer Cell Lines

The Gene Expression Profiling Interactive Analysis (GEPIA) database (source from TCGA) showed that among PRKAA1 (AMPK), SREBP2 and HMGCR, only low expression of SREBP2 in bladder cancer patients showed higher overall survival rates (Appendix A). There was no difference in PRKAA1, SREBP2 or HMGCR expression between bladder cancer tissues and adjacent normal tissues (Appendix A). Moreover, TSGH8301 cells had higher expression of p-AMPK and lower expression of SREBP2 and HMGCR than T24 cells (Appendix A). These results suggested that the decrease in SREBP2 expression might affect survival and malignancy in bladder cancers.

### 3.3. The AMPK Inhibitor Dorsomorphin Reversed the FXR Overexpression-Mediated Inhibition of Migration, Adhesion, Invasion and Angiogenesis

The AMPK inhibitor dorsomorphin was applied to cells since AMPK was activated after FXR overexpression. The wound-healing assay demonstrated that the cell migration of FXR-overexpressing TSGH8301 and T24 cells was significantly restored by approximately 60% and 25%, respectively, after treatment with dorsomorphin (Figure 2A). In the adhesion assay, the decreased adhesion in FXR-overexpressing T24 cells, but not that in TSGH8301 cells, was reversed by approximately 20% after dorsomorphin treatment (Figure 2B). In addition, treatment of FXR-overexpressing T24 cells with dorsomorphin restored invasion by approximately 35% as well as angiogenic tube length and branch point number by approximately 15% and 20%, respectively (Figure 3A,B). These results suggested that AMPK may contribute to the FXR overexpression-mediated decrease in the migration of both TSGH8301 and T24 cells and the adhesion, invasion and angiogenesis of T24 cells.

### 3.4. FXR Overexpression Attenuated Tumor Metastasis by Reducing MMP2, SREBP2, and HMGCR Expression and Enhancing p-AMPK Expression in a Tail Vein Metastatic Xenograft Model

The nude mice were injected with control or inducible FXR-overexpressing T24 cells separately twice a week for three weeks, and the mice were monitored for another three weeks to further evaluate the effect of FXR on the metastasis of human bladder cancer cells in vivo (Figure 4A). The FXR-overexpressing group exhibited a lower bioluminescence intensity in the lung than the control group after the induction of FXR expression on Days 21 and 42 according to IVIS analyses (Figure 4B). The average photon numbers (Log) decreased by almost 10 times in the FXR-overexpressing group compared to the control group (Figure 4C). Moreover, Kaplan–Meier survival curves shown that the FXR-overexpressing group (*n =* 7) had a higher survival rate than the control group (*n* = 6) (77.8% vs. 66.7%) (Figure 4D). Finally, the average body weight was not different between the CTL and FXR-overexpressing groups (Figure 4E).

The lung tumor was significantly regressed in the FXR-overexpressing group compared to the control group, as shown by HE staining (Figure 5A). According to IHC staining, the expression of MMP2, SREBP2 and HMGCR in lung tumor tissues was decreased in the FXR overexpression group compared with the control group (Figure 5B). Additionally, the protein expression of MMP2, SREBP2 and HMGCR was decreased, and the expression of p-AMPK was markedly increased in animal lung tissues (Figure 5C,D). These results showed that FXR overexpression attenuated bladder cancer metastasis and may also result in a reduction in SREBP2, HMGCR and MMP2 expression and an increase in the p-AMPK levels in the animal model.

### 3.5. The HMGCR Inhibitor Atorvastatin Enhanced the FXR Overexpression-Mediated Decreases in the Migration, Adhesion, Invasion and Angiogenesis of T24 Cells

Atorvastatin is a synthetic HMGCR inhibitor that lowers plasma cholesterol levels by inhibiting endogenous cholesterol synthesis [23]. Whether atorvastatin enhanced the effect of FXR overexpression was evaluated. The MTT assay indicated that FXR overexpression combined with atorvastatin treatment decreased cell viability by approximately 8% in T24 cells compared to FXR overexpression alone (Figure 6A). Wound-healing migration and adhesion assays demonstrated that application of atorvastatin combined with FXR overexpression showed a stronger inhibitory effect on migration in T24 cells (Figure 6B) and on adhesion in TSGH8301 cells (Appendix A).

Moreover, FXR overexpression, combined with atorvastatin treatment, enhanced the inhibition of invasion (Transwell invasion assay) by approximately 25% and angiogenesis (tube formation branch points) by approximately 10% in T24 cells (Figure 7). Application of cholesterol restored the migration and invasion, which decreased by FXR overexpression in T24 cells (Appendix A). We also found that administering cholesterol restored the tube formation of FXR-overexpressing T24 cells (Appendix A). The evidence described above suggested that the HMGCR inhibitor enhanced the FXR overexpression-mediated decreases in migration, adhesion, invasion and angiogenesis via the inhibition of cholesterol synthesis in human bladder cancer cells.

### 3.6. Clinical Effect of Statin Usage on the Overall Survival of BLCA Patients

The Kaplan–Meier estimates of the 5-year survival rate for bladder cancer patients who have taken statins versus those who have not. The log-rank test revealed a statistically significant difference between the all-cause mortality rates over time (*p* = 0.031) and bladder cancer-related mortality rates (*p* = 0.0002) (Figure 8).

Table 2 presents a summary of the Cox proportional hazards regression analyses. In multivariable risk-adjusted analysis, 5-year overall survival was significantly different between the statin used group and the non-used group in the bladder cancer cohort (adjusted hazard ratio, 0.69; 95% CI, 0.48 to 0.99, *p* = 0.043). Similarly, 5-year bladder cancer-related survival differed significantly between the statin group and the non-used group, with statin use reducing mortality by 40% (adjusted hazard ratio, 0.60; 95% CI, 0.37 to 0.96, *p* = 0.0335) among the statin use group.

The above clinical results indicated that patients who did not take statins showed a higher risk of all-cause mortality and bladder cancer-related mortality than those who were administered statins. Although the clinical data showed the benefit of statin to alleviate progression of early-stage bladder cancer, the correlation between cancer stage and cell lines was not investigated in current study. Future study was required to define the benefit of statin through FXR modulation in the aspect of tumor histology, individual cell lines and clinical cancer stage.

## 4. Discussion

FXR overexpression reduced the cholesterol contents in human TSGH8301 and T24 bladder cancer cells, and this effect may depend on the inhibition of the PI3K/AKT/mTOR signaling pathway. Activation of PI3K/AKT signaling increases intracellular cholesterol levels by inducing cholesterol synthesis via the activation of SREBP and LDL receptor-mediated cholesterol import [22,24]. In prostate cancer, AKT-mediated increases in intracellular cholesterol levels promotes cancer cell aggressiveness and bone metastasis [25,26]. Treatment with archazolid B, a chemotherapeutic drug that is in clinical bladder cancer studies, leads to the accumulation of free cholesterol within intracellular compartments by activating SREBP2 and HMGCR [27]. Archazolid B was combined with the HMGCR inhibitor fluvastatin, and the cholesterol contents and cell viability of human bladder cancer T24 cells were decreased by approximately 20% compared to archazolid B treatment alone [28]. We also treated bladder cancer cells with atorvastatin combined with FXR overexpression, which led to greater inhibition of the migration and invasion of T24 cells. Moreover, RNAseq expression in T24 control and FXR-overexpressed cells were analyzed by next-generation sequencing. When FXR was overexpressed in T24 cells, the fragments per kilobase of transcript per million (FPKM) values of HMGCR and SREBP2 were reduced (data not shown). FXR and SREBP2 has shown direct interaction and competition for transcription in human interstitial cells [29]. Therefore, the decrease of HMGCR and SREBP2 may have resulted from transcription regulation. These outcomes suggest that FXR might reduce cholesterol biosynthesis by inhibiting the PI3K/AKT/mTOR signaling pathway and subsequently decreasing the expression of SREBP2 and HMGCR in human TSGH8301 and T24 bladder cancer cells.

The AMPK inhibitor dorsomorphin restored the migration and invasion of FXR-overexpressing human bladder cancer cells. AMPK activation may act as a metabolic tumor suppressor by regulating energy levels, enforcing metabolic checkpoints and inhibiting cell growth [30]. In melanoma, AMPK activation has been shown to inhibit the metastatic potential of melanoma cells through a reduction in the activity of the ERK and COX-2 protein level [31]. In bladder cancer, sodium butyrate (NaB) induces AMPK expression, autophagy and reactive oxygen species (ROS) production, thus inhibiting the migration of human T24 and 5637 bladder cancer cells [30]. In MCF7 and MDA-MB-231 breast cancer cells, honokiol treatment increases AMPK phosphorylation and activity and the expression of its downstream targets, including acetyl-coenzyme A carboxylase and phosphorylated p70S6 kinase, which suppresses migration and invasion [32]. In melanoma, metformin has been found to inhibit cancer cell invasion, proliferation, and epithelial–mesenchymal transition through the activation of AMPK and the inhibition of MMP2 and MMP9 expression [33]. These studies imply that the increased activation of AMPK may inhibit tumor progression. In our study, we also observed that FXR overexpression decreases migration and invasion through AMPK activation.

FXR overexpression combined with atorvastatin treatment and enhanced the FXR overexpression-mediated decreases in migration, adhesion, invasion and angiogenesis in MIBC T24 cells. In colorectal adenocarcinoma, statin application may impact outcomes by decreasing invasiveness or metastatic properties and by sensitizing tumors to chemotherapeutic agents [34,35,36]. In breast cancer, a higher concentration of statins reduces the tube-forming ability of human endothelial cells in vitro by reducing the membrane localization of RhoA, subsequently suppressing VEGFR, FAK, and Akt protein expression [37]. Statins also block the adhesion, migration and invasion of breast cancer cells by reducing integrin-binding activity and MMP2 and MMP9 activity [38]. In glioblastoma, statins can modify TGF-β expression, resulting in the inhibition of migration and invasion [39]. In our results, FXR overexpression combined with atorvastatin treatment had no effect on NMIBC TSGH8301 cells. Instead, FXR overexpression combined with atorvastatin treatment enhanced the FXR overexpression-mediated decreases in migration, invasion and angiogenesis, which might have been mediated by SREBP2, HMGCR, MMP2 and VEGFA, in T24 cells. 

The clinical data showed that stage 0–I BLCA patients who were administered statins had a significantly higher overall survival rate. A retrospective population-based cohort study showed that among 13,811 individuals with NMIBC, the group of patients treated with statins (34%) after NMIBC diagnosis had a better overall survival rate but not a better cancer-specific survival rate [40]. Another prospective study showed that among 286 patients with MIBC, the 5-year LC (freedom from the development of MIBC or superficial bladder cancer necessitating cystectomy) rate of patients administered a statin compared with that of patients did not administer a statin was 73% versus 52% (*p* = 0.04) [41]. In our clinical results, stage 0–I BLCA patients who were administered statins had a significantly higher overall survival rate. Therefore, statin usage may increase patient survival in the early stage of bladder cancer.

## 5. Conclusions

The relationship between FXR expression and cholesterol biosynthesis in human bladder cancer is being investigated (Figure 9). Dorsomorphin treatment reversed, while atorvastatin treatment enhanced, the FXR overexpression-mediated inhibition of migration, invasion and angiogenesis. Furthermore, FXR overexpression decreased MIBC T24 cell metastatic ability by downregulating MMP2, SREBP2 and HMGCR expression in vivo. These findings may indicate that FXR overexpression combined with atorvastatin treatment may be a potential therapeutic strategy for the treatment of human bladder cancer in the future.

## Figures and Tables

**Figure 1 cancers-14-04398-f001:**
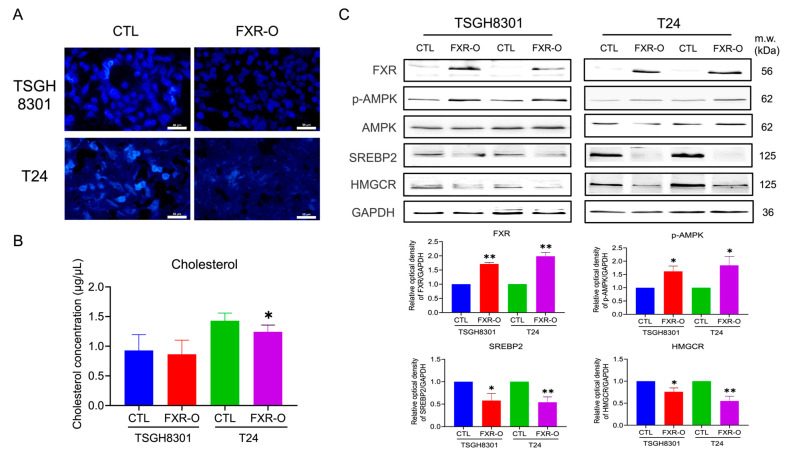
The effect of Farnesoid X receptor (FXR) overexpression on cholesterol content and cholesterol biosynthesis-related protein expression. (**A**) FXR was overexpressed (FXR-O) in TSGH8301 and T24 cells for 72 h, and the cholesterol content was measured by a cholesterol cell-based assay. (**B**) The level of cholesterol secreted into the conditioned medium of TSGH8301 and T24 cells was analyzed by ELISA. (**C**) FXR was overexpressed in TSGH8301 and T24 cells for 72 h, and p-AMPK, AMPK, SREBP2 and HMGCR protein expression was examined by Western blotting. GAPDH was a loading control. The relative quantitative analysis of these proteins was shown by bar graphs. * *p* < 0.05; ** *p* < 0.01 compared with the control group. Scale bar = 50 μm. The original blots are shown in Appendix A.

**Figure 2 cancers-14-04398-f002:**
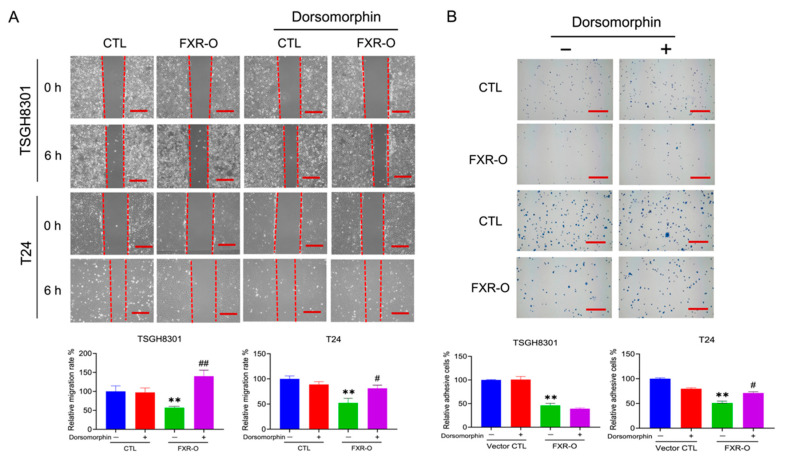
The effect of the AMPK inhibitor dorsomorphin on migration and adhesion in human bladder cancer cells after FXR overexpression. (**A**) The migratory ability of FXR-overexpressing (FXR-O) TSGH8301 and T24 cells treated with or without dorsomorphin (5 μM) was analyzed by wound healing assay after scratching for 6 h. The right panel shows the relative migration rate. (**B**) The adhesive ability in FXR-O TSGH8301 and T24 cells treated with or without dorsomorphin was analyzed after 50 min of incubation. Then, the adherent cells were stained and photographed. The right panel shows the relative numbers of adherent cells. ** *p* < 0.01 compared to the control group. # *p* < 0.05; ## *p* < 0.01 compared to the FXR-O group. Scale bar = 200 μm.

**Figure 3 cancers-14-04398-f003:**
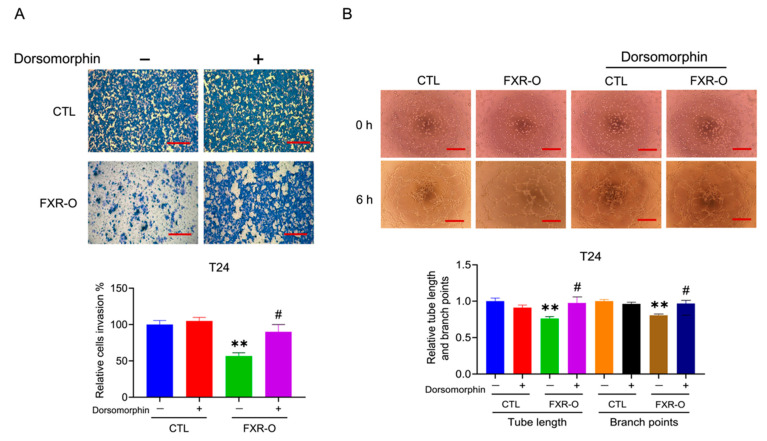
The effect of dorsomorphin on invasion and angiogenesis of human bladder cancer T24 cells after FXR overexpression. (**A**) The invasive ability of muscle-invasive FXR-O T24 cells treated with or without dorsomorphin was analyzed by Transwell assay with Matrigel for 18 h. The cells were stained and imaged. The lower panel shows the quantitative results. (**B**) HUVECs were cultured with control or FXR-O conditioned medium with or without dorsomorphin for 6 h. The number of branch points and lengths of endothelial cell networks were analyzed. The quantitative results of relative branch point numbers and tube lengths were shown as below. ** *p* < 0.01 compared to the control group; # *p* < 0.05 compared to the FXR-O group. Scale bar = 200 μm.

**Figure 4 cancers-14-04398-f004:**
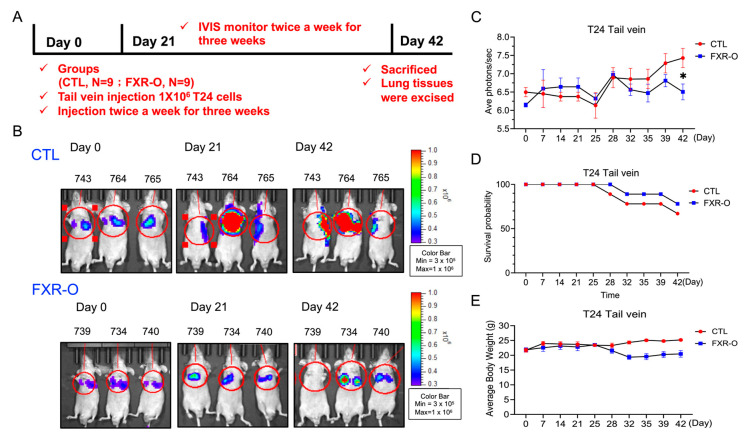
The effect of FXR overexpression on lung metastasis in a xenograft mouse model. (**A**) Experimental timeline. (**B**) The effects of control and FXR-O on the size and growth of tumors were observed using bioluminescent imaging with an in vivo imaging system (IVIS). (**C**) The luminescence intensity of photons emitted from each tumor was quantified (Log). (**D**) The animal survival rate was observed over the 42-day experimental period shown by a Kaplan-Meier survival curves. (**E**) Body weights were measured during the experimental period. * *p* < 0.05 compared to the control group.

**Figure 5 cancers-14-04398-f005:**
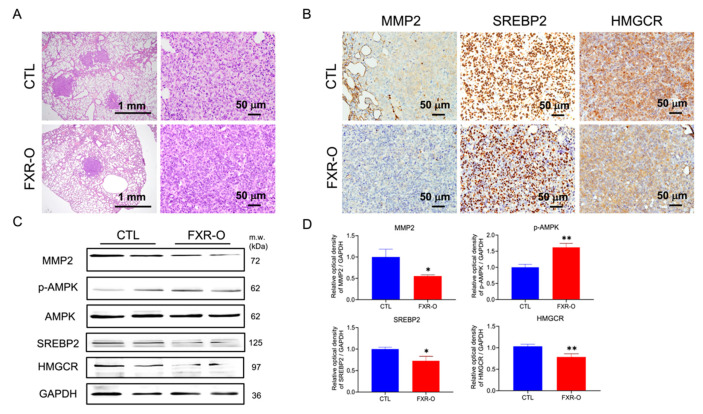
The effect of FXR overexpression on cholesterol biosynthesis-related protein expression in lung tissues of the xenograft model. (**A**) Lung tumors were observed by hematoxylin and eosin staining. Scale bar = 1 mm and 50 μm. (**B**) Immunohistochemical analysis of MMP2, SREBP2 and HMGCR expression in lung tissues from the control and FXR-O groups. Scale bar = 50 μm. (**C**) MMP2, p-AMPK, AMPK, SREBP2 and HMGCR protein expression was measured by Western blotting analysis. (**D**) The relative quantitative analysis of these proteins was shown by bar graphs. * *p* < 0.05; ** *p* < 0.01 compared to the control group. The original blots are shown in Appendix A.

**Figure 6 cancers-14-04398-f006:**
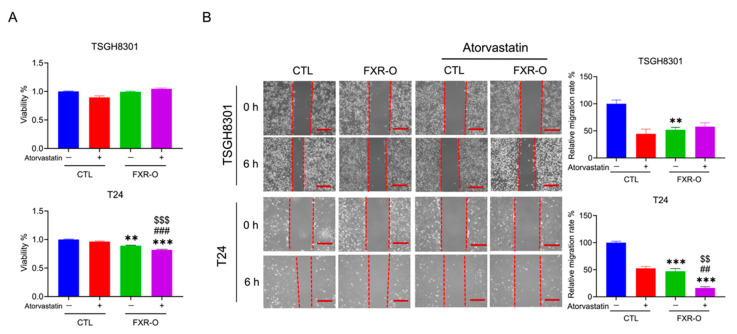
The effect of FXR overexpression combined with atorvastatin treatment on survival and migration. (**A**) The viability of FXR-O TSGH8301 and T24 cells treated with or without atorvastatin (20 µM) was examined by MTT assays for 24 h. (**B**) Wound healing migration assays were performed in control and FXR-O TSGH8301 and T24 cells treated with or without atorvastatin. The lower panel shows the quantified relative migration rate. ** *p* < 0.01; *** *p* < 0.001 compared to the control group. ## *p* < 0.01, ### *p* < 0.001 compared to the control + atorvastatin group. $$ *p* < 0.01; $$$ *p* < 0.001 compared to the FXR-O group. Scale bar = 200 μm.

**Figure 7 cancers-14-04398-f007:**
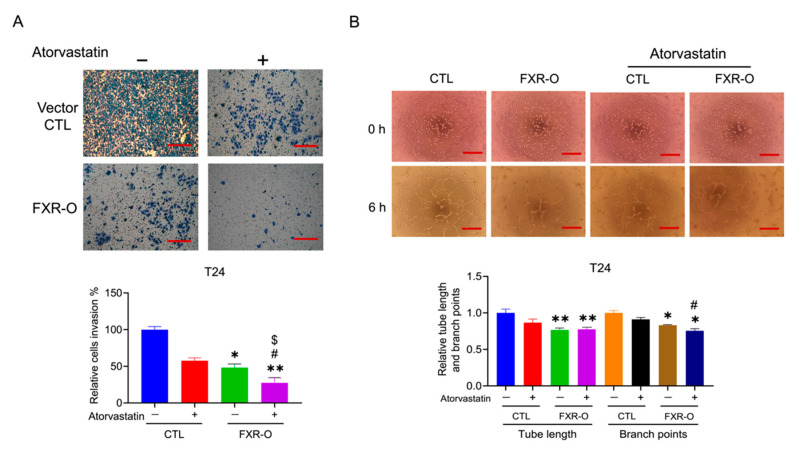
The effect of FXR overexpression combined with atorvastatin treatment on the invasion and angiogenesis of T24 cells. (**A**) Transwell invasion assays were performed in control and FXR-O T24 cells treated with or without atorvastatin. The lower panel shows the relative migration rate. (**B**) Tube formation assays were performed in control and FXR-O T24 cells treated with or without atorvastatin. The number of branch points and tube length of endothelial cell networks were analyzed. The quantitative results of the analysis of relative branch point numbers and tube lengths were shown below. * *p* < 0.05; ** *p* < 0.01; compared to the control group. # *p* < 0.05 compared to the control + atorvastatin group. $ *p* < 0.05 compared to the FXR-O group. Scale bar = 200 μm.

**Figure 8 cancers-14-04398-f008:**
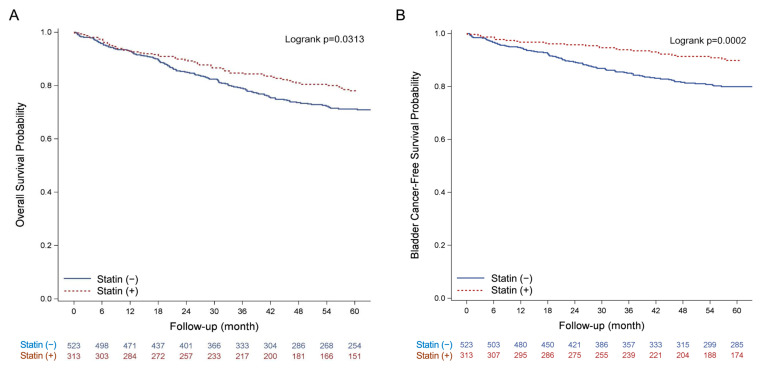
Kaplan–Meier plot of stage 0–I bladder cancer mortality with statins versus without. (**A**) all-cause mortality; (**B**) bladder cancer-related mortality.

**Figure 9 cancers-14-04398-f009:**
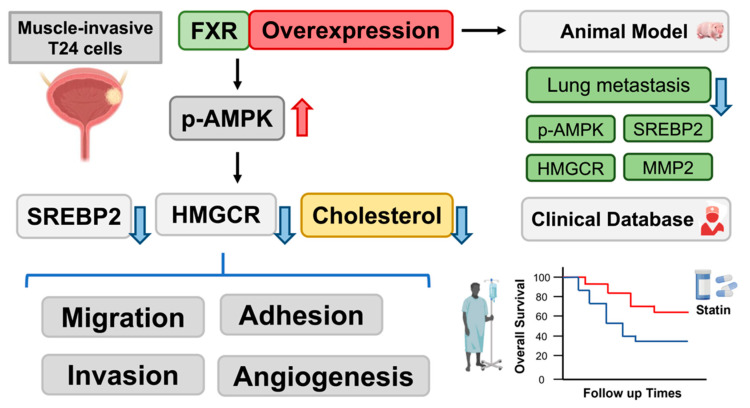
Scheme of FXR overexpression effects in vitro, in vivo and statin usage with BLCA patients in clinical database.

**Table 1 cancers-14-04398-t001:** Characteristic of BLCA stage 0-I patients (*n* = 836).

Variable	Statins Use of Patients (%)		*p*-Value
	Statin (+) (*n* = 313)	Statin (−) (*n* = 523)	
Age, Mean (SD)	69.6 (12.6)	68.4 (14.0)	0.22
Gender			0.42
Male	236 (75.4)	407 (77.8)	
Female	77 (24.6)	116 (22.2)	
T category			0.53
T1	313 (100.0)	521 (99.6)	
T2	0 (0.0)	2 (0.4)	
N category			-
N0	313 (100.0)	523 (100.0)	
Cell type			0.0017
Urothelial carcinoma	296 (94.6)	455 (87.0)	
Adenocarcinoma, Squamous carcinoma, Neuroendocrine carcinoma	3 (1.0)	7 (1.3)	
Others	14 (4.5)	61 (11.7)	
BMI, kg/m			<0.0001
<25	167 (53.4)	381 (72.8)	
≥25	146 (46.6)	142 (27.2)	
CCI			<0.0001
0	142 (45.4)	410 (78.4)	
1–2	92 (29.4)	74 (14.1)	
3+	79 (25.2)	39 (7.5)	
Lipid profiles, mg/dL			
Total Cholesterol, mg/dL	206.9 (44.8)	182.3 (33.5)	<0.0001
Normal or low (<200)	82 (26.2)	59 (11.3)	
High (≥200)	105 (33.5)	26 (5.0)	

Abbreviation: BMI body mass index, CCI Charlson Comorbidity Index, HDL-C high-density lipoprotein cholesterol, LDL-C low-density lipoprotein cholesterol, SD standard deviation.

**Table 2 cancers-14-04398-t002:** Cox proportional-hazards model of mortality among stage 0–I bladder cancer patients with and without statins use.

	No.	Events (%)		Crude HR (95% CI)		*p*-Value	Adjusted ^†^ HR (95% CI)		*p*-Value
**All-cause mortality**									
Statin (−)	523	134	(25.6)	1.00	(Ref.)		1.00	(Ref.)	
Statin (+)	313	60	(19.2)	0.72	(0.53–0.97)	0.0321	0.69	(0.48–0.99)	0.0433
**Bladder cancer-related mortality**									
Statin (−)	523	93	(17.8)	1.00	(Ref.)		1.00	(Ref.)	
Statin (+)	313	27	(8.6)	0.46	(0.30–0.70)	0.0003	0.60	(0.37–0.96)	0.0335

^†^ Adjusted for age, gender, cell type, BMI, CCI, TC, LDL-C. Abbreviations: BMI, body mass index; CCI, Charlson Comorbidity Index; CI, confidence interval; HR, hazard ratio; LDL-C, low-density lipoprotein cholesterol; TC, total cholesterol.

## Data Availability

The data presented in this study are available on request from the corresponding author.

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
