# Peer review of "Farnesoid X Receptor Overexpression Decreases the Migration, Invasion and Angiogenesis of Human Bladder Cancers via AMPK Activation and Cholesterol Biosynthesis Inhibition"

_cancers, 2022, doi:10.3390/cancers14184398_

Round 1

Reviewer 1 Report

This is an interesting manuscript and providing important information. The following are some comments.

1.Since the clinical data suggested that patients with stage 0-1 bladder cancer had better OS if they took statins. It will be better to include a proposed mechanism shy FXR overexpression combined with atorvastatin had effects on the T24 cell line, but not NMIBC 477 RSGH8301 cells.

2. In table 1 , why the cut-off value of BMI was 25. Is there any relevant reference?

3. The clinical data that the authors provided indicated that the group taking  statin had statistically significant better CCI than those who did not take statins. Therefore, the better all cause survival may due to better baseline health condition but not the effect of drug. Is this a possibel confounding factor? 

Author Response

Reply to Reviewer 1’s comments

This is an interesting manuscript and providing important information. The following are some comments.

1.Since the clinical data suggested that patients with stage 0-1 bladder cancer had better OS if they took statins. It will be better to include a proposed mechanism shy FXR overexpression combined with atorvastatin had effects on the T24 cell line, but not NMIBC TSGH8301 cells.

Reply: Thank you for your viewpoint. Our clinical data showed the benefit of statin to alleviate progression of early-stage bladder cancer. The in vitro results delineated the possibility of FXR involvement in statin advantage in bladder cancer. The correlation between cancer stage and cell lines was not investigated in current study. We had added supplementary description in the manuscript to address this point. Future study was required to define the benefit of statin through FXR modulation in the aspect of tumor histology, individual cell lines and clinical cancer stage.

Page 12, line 441

Although the clinical data showed the benefit of statin to alleviate progression of early stage bladder cancer, the correlation between cancer stage and cell lines was not investigated in current study. Future study was required to define the benefit of statin through FXR modulation in the aspect of tumor histology, individual cell lines and clinical cancer stage.

  1. In table 1, why the cut-off value of BMI was 25. Is there any relevant reference?

Reply: Thank you for the comment. In this study, we used the National Heart, Lung, and Blood Institute's terminology with categories of normal weigh (BMI of <25) [1]. And we appreciate the reviewer's reminder. We realized we made a minor mistake when we set the equal sign to less than or equal to 25. The corrected number of cases remains unchanged.

The following are sources for reference: Flegal KM, Kit BK, Graubard BI. Body mass index categories in observational studies of weight and risk of death. Am J Epidemiol. 2014 Aug 1;180(3):288-96. doi: 10.1093/aje/kwu111. Epub 2014 Jun 3. PMID: 24893710; PMCID: PMC4732880.

  1. The clinical data that the authors provided indicated that the group taking statin had statistically significant better CCI than those who did not take statins. Therefore, the better all cause survival may due to better baseline health condition but not the effect of drug. Is this a possible confounding factor? 

Reply: Thank you for the reviewer’s insightful comments. Differences in co-morbidity between the two groups are likely to confound the study results. As the clinical database is an observational study, it is hard to anticipate interference variables as in a randomized controlled trial. In this study, we attempted to adjust for demographic differences using other methods such as propensity score matching or IPTW weighting; however, despite the adjustments, there was still an imbalance between the two CCI groups due to the large variance between them. However, the severity of CCI among statin users in this study was greater than among non-users. Thus, we finally also utilized multivariate analysis and modeling to account for potential confounding variables, such as CCI. Per our study, patients with bladder cancer who took statins had a lower mortality rate from all causes and from bladder cancer, despite having more co-morbidities.

Reference

  1. Flegal KM, Kit BK, Graubard BI: Body mass index categories in observational studies of weight and risk of death. Am J Epidemiol 2014, 180(3):288-296.

Reviewer 2 Report

Farnesoid X receptor overexpression decreases the migration, invasion and angiogenesis of human bladder cancers via AMPK activation and cholesterol biosynthesis inhibition

Chien-Rui Lai, Yu-Ling Tsai, Wen-Chiuan Tsai, Tzu-Min Chen, Hsin-Han Chang, Chih- Ying Changchien, Sheng-Tang Wu, Hisao-Hsien Wang, Ying Chen, Yu-Huei Lin

Summary:

Lai et. al. describe the overexpression of FXR and its effects on cholesterol metabolism through regulation of HMGCR, SREBP and AMPK proteins.

Major questions/concerns:

1.    Dorsomorphin is a general AMPK inhibitor which likely has secondary effects through the mTOR complex. To complement the results and observations put forth in this study (Fig. 2 and Fig.3), the chemical inhibition studies need to be complemented with genetic perturbation studies – knocking down/out the genes involved.

2.    What is the rationale for using the two cell lines in this study? Why are these cell lines good systems to test the FXR-cancer progression axis?

3.    For Figs. 2, 3, 6, and 7, including the data for time 0 (or an early time point) as a control will provide a better appreciation of the effects being observed.

4.    There are very few, if any at all, experiments to prove that FXR overexpression dampens the progression of bladder cancers via reduced cholesterol (except for Fig. 1A). This measure should be included in all the experiments performed. Fig. 4 could include a panel of measurements of serum cholesterol from the animals (at least the serum cholesterol levels on Day 42).

5.    SREBP is a transcription factor with HMGCR being one of the transcriptional targets. Upon overexpression of FXR, are the levels of SREBP and HMGCR are both reduced. Is this because of the reduced SREBP mediated transcription alone? Or, are there other metabolic changes happening in the cell that lead to degradation of HMGCR in response to FXR overexpression, in addition to, and perhaps independent of, transcriptional down regulation?

Minor comments/concerns:

Line 28: “Additionally, Statis clinical data showed….” Is there a typo?

Line 76 (and Line 463): Several papers do exist that talk about the relationship between FXR and cholesterol metabolism!

Sec 2.2: Please reword/rewrite to make this section succinct. The NM number of the gene cloned in, the vector back bone used, sites and/or method of cloning, etc.

Line 202: “… the information has been collected from 1997. …” Do you mean ‘since (the year) 1997’ or ‘over the past 25 years’ ?

Lines 208-209: I did not understand what it means/trying to say. Can you please edit to explain better?

In discussing Fig. 1, it would be nice to compare what percentage of total AMPK is phosphorylate, and what that may imply that overexpression of FXR in turn results in a change in the fraction of AMPK that is phosphorylated.

To present the data in Fig. 4D, a Kaplan-Meier survival curve would be a more appropriate way.

Line 329: Is it VEGF or VEGFR1 that is being talked about. Fig S3 shows blots for VEGFR1 and VEGFR2.

Lines 351-353: It sounds like a circular argument, and hence it was a little confusing to read and understand. The language could be simplified to succinctly put forth the argument being made.

Figure 9 needs to be refined, so as to show the interplay between AMPK and PI3K/AKT/mTOR; and to simplify what is being depicted.

Reviewer 3 Report

Dear authors, unfortunately the message, experimental development and conclusions obtained are not sufficiently impressive and attractive for future readers of Cancers. I appreciate the work done and I wish luck in new attempts in other journals

   Best Regards

Author Response

Reply to Reviewer 3’s comment

Dear authors, unfortunately the message, experimental development and conclusions obtained are not sufficiently impressive and attractive for future readers of Cancers. I appreciate the work done and I wish luck in new attempts in other journals.

Reply: We really appreciated for the comments. This is the first studies to figure out that upregulation of FXR inhibited tumor progression in urothelial carcinoma with AMPK activation and cholesterol reduction. In addition, we conducted not only in vitro muscle invasive and non-muscle invasive cell lines and in vivo nude mice animal model but also obtained the clinical statistics with or without statin usage of patients and overall survival rate follow up for 60 months. All the above evidence indicated that FXR overexpression combined with statin usage may strengthen the inhibition ability of migration, invasion and angiogenic in bladder cancer. Chenodeoxycholic acid (CDCA), an FXR agonist, is provided for the cerebrotendinous xanthomatosis patients. CDCA is approved by FDA and might suppress the activity of 3-hydroxy-3-methylglutaryl coenzyme A (HMG CoA) reductase in the liver [1]. In conclusion, our studies may provide the newly therapeutics strategies by combining statin and FXR agonist like CDCA in human bladder cancers. And it may have strong potentials to give another option in clinical application in the future.

Reference

  1. Fiorucci S, Distrutti E: Chenodeoxycholic Acid: An Update on Its Therapeutic Applications. Handb Exp Pharmacol 2019, 256:265-282.

Round 2

Reviewer 1 Report

The authors could answer questions and revised accordingly. 

Author Response

Thank you for the reviewing and giving us valuable comments.

Reviewer 2 Report

Major Comment 1:
The western blots - especially the control GAPDH - are so poorly run, that I am not entirely sure it would be appropriate to use that.

Major Comment 2 (and Minor comment 2):
There are several papers that dwell on the FXR-Cholesterol axis in human bladder cancers. Although, at this point, just semantics, it is important to cite relevant literature. Indeed, the authors in reply to major comment #2 state that "This study is a continuation of our previous study in bladder cancers."

Quoted from a previous paper - from May 2022 - by the same authors "The farnesoid X receptor (FXR, encoded by the NR1H4 gene) functions as a bile acid nuclear receptor [7], and is expressed mainly in the liver, intestine, kidney, and adrenal glands [8]. After being activated by ligands, FXR interacts with its heterodimer partner retinoid X receptor (RXR) and binds to FXR response element (FXRE). Next, FXR modulates the expression of the downstream target genes, including bile acid homeostasis [9], fatty acid metabolism [10], and glucose metabolism [11]. Moreover, FXR induces gene expression of small heterodimer partner (SHP), which suppresses the expression of CYP7A1 and reduces hepatic bile acid synthesis via negative feedback in cholesterol metabolism [12]."

Int J Mol Sci. 2022 May; 23(9): 5259. Published online 2022 May 9. doi: 10.3390/ijms23095259

Major Comment 4:
There are commercially available ELISA kits which could be used to test the levels of cholesterol. And/or common lab tests can be performed - at least attempted - to determine the levels of cholesterol.

Major Comment 5:
This information should be included in the discussion and/or the relevant results sections.
  Minor Comment 9:
To say that overexpression of FXR and atrovastatin work synergistically, each of their individual effects needs to shown - which I don't think has been done in this study?!
